# How Many Messenger RNAs Can Be Translated by the START Mechanism?

**DOI:** 10.3390/ijms21218373

**Published:** 2020-11-08

**Authors:** Laurence Despons, Franck Martin

**Affiliations:** Institut de Biologie Moléculaire et Cellulaire, “Architecture et Réactivité de l’ARN” CNRS UPR9002, Université de Strasbourg, 2, allée Konrad Roentgen, F-67084 Strasbourg, France; despons@unistra.fr

**Keywords:** ribosome, translation initiation, START, secondary structures, mRNA

## Abstract

Translation initiation is a key step in the protein synthesis stage of the gene expression pathway of all living cells. In this important process, ribosomes have to accurately find the AUG start codon in order to ensure the integrity of the proteome. “Structure Assisted RNA Translation”, or “START”, has been proposed to use stable secondary structures located in the coding sequence to augment start site selection by steric hindrance of the progression of pre-initiation complex on messenger RNA. This implies that such structures have to be located downstream and at on optimal distance from the AUG start codon (i.e., downstream nucleotide +16). In order to assess the importance of the START mechanism in the overall mRNA translation process, we developed a bioinformatic tool to screen coding sequences for such stable structures in a 50 nucleotide-long window spanning the nucleotides from +16 to +65. We screened eight bacterial genomes and six eukaryotic genomes. We found stable structures in 0.6–2.5% of eukaryotic coding sequences. Among these, approximately half of them were structures predicted to form G-quadruplex structures. In humans, we selected 747 structures. In bacteria, the coding sequences from Gram-positive bacteria contained 2.6–4.2% stable structures, whereas the structures were less abundant in Gram-negative bacteria (0.2–2.7%). In contrast to eukaryotes, putative G-quadruplex structures are very rare in the coding sequence of bacteria. Altogether, our study reveals that the START mechanism seems to be an ancient strategy to facilitate the start codon recognition that is used in different kingdoms of life.

## 1. Introduction

In all living organisms, translation is a universal mechanism that enables the biosynthesis of proteins according to the nucleic acid-encoded genetic information. The platform that allows the sequential assembly of amino acids is the ribosome, a macromolecular machine found in all three kingdoms of life. Translation consists of four steps: translation initiation, elongation, termination and ribosome recycling. Among these, the rate-limiting step is, most of the time, translation initiation and is therefore considered as the master regulator step, although there are examples where the elongation rate can dominate. In this initiation phase, the ribosome has to accuracy localize the start codon, most frequently encoded by an AUG triplet. Importantly, the cell integrity depends directly on the efficiency and the fidelity of this critical step. Therefore, living organisms have evolved sophisticated molecular mechanisms that ensure proper translation initiation in order to avoid aberrant protein synthesis. These mechanisms have been extensively studied in the last three decades in prokaryotes [1,2], in eukaryotes [3,4,5,6,7,8,9,10] and in archaea [11]. Interestingly, rapid cellular adaptation is achieved essentially by post-transcriptional events. Among these, localization of the start codon is a critical step that is not only essential for cell maintenance, it is also essential to modify the proteome. For instance, in prokaryotes, the proteome is changed in order to adapt quickly to new environmental conditions. In eukaryotes, the translational landscape can be drastically modified to allow fast and efficient stress response to various stimuli (infection, nutrient deprivation, developmental stages, etc.). Localization of the AUG start codon can be a challenging task due to various obstacles. For instance, mRNAs have the propensity to fold and form stable secondary structures such as stem-loops [12], tertiary structures [13,14] or more sophisticated structures termed G-quadruplexes [15]. Translation initiation efficiency is greatly influenced by the presence of secondary structures in the 5′UTR. In addition, it is well established that structures directly at or in close proximity to a start codon are inhibitory because they prevent efficient ribosome loading [16]. However, in the case of internal ribosome entry sites (IRES), structures located close to the start codons are usually crucial for efficient translation [17]. 

In prokaryotes, the ribosome binding site (RBS) that encompasses the “Shine-Dalgarno” (SD) sequence must be accessible and not involved in structures to allow efficient translation initiation. During initiation, the 3′ end of the 16S ribosomal RNA anneals with the SD sequence on the mRNA in order to ensure efficient assembly of the pre-initiation complex. Indeed, hindering of the RBS in a structure is an efficient way to down-regulate translation initiation. In fact, these structures are considered as genuine switches that can turn on translation by unwinding them, thereby liberating the RBS for efficient translation initiation. Many examples of such structures in the RBS leading to SD sequence hindering have been described [18,19,20]. The largest acidic ribosomal protein S1 is required for efficient unfolding of RNA duplexes hindering the SD sequence [21,22]. The folding/unfolding of these structures is also controlled by various mechanisms including *cis*-acting elements such as riboswitches that can sense several biophysical parameters like temperature, pH or the presence of specific ligands or metabolites or regulatory RNAs or proteins [23]. Indeed, translation initiation requires the anchoring of the ribosomal RNA on the SD sequence. Nowadays this point is very well documented ([1] and references therein). Nevertheless, a recent genome-wide analysis revealed that a large fraction (15–100%) of prokaryotic transcripts is translated by another mechanism that is SD-independent. These mRNAs have no 5′UTR (leaderless mRNAs) or do not contain any SD-like sequences [24]. These SD-independent translation events occur in bacterial and organellar genomes. Moreover, in the entire Gram-negative bacterial phylum Bacteroidetes, the ribosome does not use SD interactions to initiate translation [25]. Apart from the fact that the 5′UTR of these mRNAs is usually unfolded, the molecular basis of such an SD-independent translation initiation mechanism remains currently elusive [24].

In eukaryotes, the 5′UTRs are generally much longer and consequently more likely to harbor stable secondary structures. The median length of 5′UTRs in eukaryotes ranges from 53 nucleotides in yeast to 218 nucleotides in humans [26,27,28,29,30]. First, an mRNA activation step triggers the assembly of the eIF4F complex on the 5′ cap. This complex contains eIF4E, the cap binding protein, the RNA helicase eIF4A and its auxiliary factor eIF4B and the platform protein eIF4G. Then, the assembly of the translation machinery occurs at the 5′ cap, and a so-called scanning step enables the 5′-3′ sliding of the 43S particle (which contains the 40S ribosomal subunit, the multi subunit factor eIF3, the scanning factors eIF1, 1A and 5 and the so-called ternary complex (comprising eIF2 and the initiator tRNA^Met^)) in order to localize the start codon. This scanning step has been extensively described by detailed mechanistic and functional studies [3,31] but has been effectively observed only very recently [32,33]. The presence of these stable structures in both prokaryotes and eukaryotes requires a specific mechanism in order to enable proper access to the AUG start codon. Specific RNA helicases that can unwind such structures are found in prokaryotes [34,35] and in eukaryotes [36]. 

Stable secondary structures can also be found downstream of the start codon in the coding sequence. These sequences, when they are located at the proper distance from the start codon, can influence translation initiation efficiency both in prokaryotes [37] and in eukaryotes [38]. High-throughput methods to find secondary structures in early coding sequences have been conducted [39,40,41,42]. These studies converged on the finding that the region around the start codon of a great majority of mRNAs is, most of the time, poorly structured. 

Here we performed genome-wide analyses of the beginning of the coding sequences in order to identify putative secondary structures that might influence translation initiation efficiency. Indeed, the presence of stable structures, when located downstream at the optimal distance from the AUG start codon, can greatly facilitate translation initiation by a novel mechanism that we named the Structure Assisted RNA Translation (START) mechanism [38]. More precisely, the START mechanism relies on the presence of stable secondary structures located in the coding region. These structures are located at the appropriate distance from the start codon (+16 to +19) that enables the assembly of the pre-initiation complex on the AUG start codon without unfolding of the structures. In fact, these structures at this position slow down, by steric hindrance, the progression of the scanning pre-initiation and thereby augment the initiation efficiency. Thus, their precise position downstream +16 to +19 is crucial for efficient localization of the start codon.

In order to evaluate the importance of this START mechanism in the cell, we underwent a genome-wide screening of secondary structures located in the coding sequence downstream of the AUG. We screened six eukaryotic genomes and eight prokaryotic genomes. 

## 2. Results and Discussion

During protein biosynthesis, the ribosome slides 5′ toward 3′ on the mRNA and orchestrates the sequential addition of amino acids by recruiting the cognate aminoacyl-tRNAs accordingly to the codon sequence. Various biochemical and structural studies have shown that the fully assembled ribosome covers a region encompassing approximately 25 to 30 nucleotides of the messenger RNA [43,44] (Figure 1A). In the initiation step, the ribosome is positioned with the AUG codon in its P-site. Traditionally, the initiating ribosome is located on the mRNA by the so-called “toe-printing” experiment. The principle of this method is to map the position of the ribosome on the mRNA in the initiation phase, thanks to a primer extension assay that leads to a reverse transcriptase arrest, called a “toe-print”, induced by the presence of the ribosome. Indeed, the position of the “toe-print” corresponds to the 3′ edge of the ribosome on the mRNA. By convention, the A of the AUG start codon is numbered +1. During translation initiation, the ribosome is positioned on the AUG start codon in its P-site. Toe-printing experiments performed by many researchers over the last three decades have shown that when the ribosome is positioned on the AUG, the typical “toe-print” is usually detected at position +15 to +17 (Figure 1A). Recently, we have described a novel translation mechanism that we named “Structure Assisted RNA Translation”, or START [38]. The mRNAs that are translated by this mechanism harbor a stable secondary structure that is located downstream nucleotide +19. By steric hindrance, these structures assist the ribosome to accurately localize the AUG start codon and therefore improve the translation efficiency of these mRNAs [38] (and references therein). In order to evaluate how many mRNAs are translated by the START mechanism, we performed genome-wide screenings for the presence of putative secondary structures located in this critical area on six eukaryotic organisms and eight prokaryotic organisms. The screenings were performed on the coding sequences, and we targeted our search in a window of 50 nucleotides. We focused our efforts in order to find structures located at the optimal distance required for the START mechanism, precisely from +16 to +65 (Figure 1B). The strategy was to detect putative secondary structures by calculating the minimal free energy (MFE) of the folded structure. In order to select significant structures, we proceeded as follows. In the first step, we calculated the MFE of a putative structure (MFE_struc_). Then in the second step, the corresponding selected sequence was randomized by mixing the nucleotides present in this 50 nt window. This randomization was repeated 100 times for each sequence, and we calculated the MFE of the resulting randomized sequence (MFE_rand_). Only the structures with an MFE_struc_/MFE_rand_ ratio equal or higher than 2 were retained. An additional criterion for selecting the structures was used. We retained the top 25% of the structures, i.e., the structures that had an MFE_struc_ equal or lower than the value of the quantile 0.25. This value was calculated from the distribution of MFE_struc_ values of all structures of a given species. Among the selected structures, we also predicted the presence of putative G-quadruplex structures. These G-quadruplex structures are formed by the stacking of so-called G-quartets composed by 8, 12, 16, 20 G or even more G residues (Figure 1C).

In eukaryotic coding sequences, we found a significant number of structures (Appendix A). For example, in humans 747 structures have been retained (Figure 2A). Among these, an important part is composed by G-quadruplexes, although the ratio is variable from one to another organism. Indeed, almost half of the detected structures are G-quadruplexes in humans and mice, whereas G-quadruplexes are rare in *Caenorhabditis elegans*. Remarkably, G-quadruplexes are almost absent in *Saccharomyces cerevisiae*. However, the proportion of mRNAs containing structures are higher in yeast reaching ~2.5% of the total coding sequences. This proportion is almost as high as in yeast in *C. elegans* (~2%) but is significantly lower in the human and mouse (less than 1%) (Figure 2B).

We performed the same analysis on prokaryotic species. We screened the coding sequences from four Gram-positive bacteria and four Gram-negative bacteria (Appendix A). In this case, the diversity of the result of our screening is much higher (Figure 3A). We found bacteria that contain many structures like *Bacillus anthracis* with 167 structures and bacteria that contain very few structures, such as *Pseudomonas aeruginosa* with only 10 structures. Concerning the type of structures, G-quadruplexes are rare in all the bacteria tested. However, 168 and 161 G-quadruplexes have been found in *Escherichia coli* and *P. aeruginosa,* respectively, mainly in the coding sequences [45], indicating that G-quadruplexes are also present in bacteria but very rarely in the +16 to +65 window. Altogether, our data seem to indicate that the appearance of such structures in this area of the coding sequence is rather a eukaryotic feature. In agreement with the observation that G-quartets are pretty common in the coding sequences of eukaryotes, specific RNA helicases dedicated to G-quadruplexes such as DHX36 and DHX9 have been found in eukaryotic genomes [46]. In addition, in humans G-quadruplexes are not always present because they are only transiently folded under specific conditions [47]. This suggests that the START mechanism would be activated only in specific conditions that would induce the formation of G-quadruplexes in the +16 to +65 window, thereby adding another layer of regulation in the translation process. 

Interestingly, the proportion of structures selected is in general much lower in Gram-negative bacteria compared to Gram-positive bacteria, suggesting that START is more common in Gram-positive bacteria (Figure 3B). Translation initiation in Gram-positive bacteria is far less understood than in Gram-negative bacteria. Indeed, the presence, the nature and the use of the SD sequence are features that remain rather poorly studied in Gram-positive bacteria. Since the importance of the SD sequence is lower, the START mechanism might be another alternative used for translation initiation, which might explain that more structures are found by our screen in Gram-positive bacteria. 

In prokaryotes, the transcription process has been shown to be coupled to translation. The time needed to fold a hairpin helix is in the millisecond range [48,49]. Translation initiation frequency by the ribosome is rather in the second range [50]. Therefore, secondary structures in the coding region have enough time to fold before translation initiation occurs, which indicates that the START mechanism is indeed possible even in prokaryotes when transcription and translation are coupled. 

Next, we examined the possibility that the differences we observed in the different organisms were actually reflecting the GC content of the coding sequences from each genome. Thus, we wondered whether a high GC content in the coding sequences would be more prone to structures and favor the START mechanism. In the eukaryotes tested, we observed that the GC content was very similar, around 50% in all the genomes analyzed here, the lowest GC content being for yeast with 41.66% (Figure 4A). This is an interesting point because yeast is the eukaryotic organism that contains the highest proportion (2.5%) of structures in the beginning of its coding sequence, indicating that there is no correlation between the GC content and the number of selected secondary structures. By calculating the MFE for each structure, we can also evaluate the stability of these structures. We investigated the stability of the top 25% of the structures found in each organism (Figure 4B). The more stable structures were found in humans and mice, while the less stable were found in yeast and *C. elegans*. When looking at the stability of the selected structures, there was a perfect correlation with the GC content. This is rather expected since mice and humans also contain the highest numbers of G-quadruplexes. This suggests that G-quadruplexes are generally more stable than other types of secondary structures.

In prokaryotes, the situation is significantly different. The GC contents of Gram-negative bacteria were generally higher than those in Gram-positive bacteria, the extreme example being *P. aeruginosa* (65.41%) (Figure 5A). Like in eukaryotes, there was a perfect correlation between the GC content and the stability of the top 25% of the structures (Figure 5B). Nevertheless, the Gram-negative bacteria do contain fewer structures than Gram-positive bacteria, as previously mentioned. This is particularly striking for *P. aeruginosa*, the bacteria with the highest GC content and the least structures selected (only 10 structures selected). Another interesting point is the presence of a lot of structures in Gram-positive bacteria, however with a medium stability (Figure 5B). In conclusion, the GC content is a good predictor for the presence of structures in eukaryotic coding sequences but not for prokaryotic coding sequences. This is probably due to the high occurrence of G-quadruplexes, a eukaryotic feature as previously mentioned.

Next, we analyzed the types of mRNA that contained the selected structures by examining their GO terms (Appendix A). In eukaryotes, no specific biological process was enriched among the selected mRNAs. In contrast, in bacteria, the translation process was the first hit for all the bacteria tested here. Therefore, according to the structures that were selected, the START mechanism seems to be particularly important for translation regulation in bacteria. When looking at the functions of the selected mRNAs, the first hits for all the organisms tested were ATP binding or metal ion binding. Concerning the localization, the selected mRNA code, in general, for proteins were almost equally distributed in all the compartments of the cell. 

Finally, we had a deeper look at the most stable structures that we found in human mRNAs (Figure 6 and Figure 7, Appendix A). The stability of these structures ranged from −41.8 to −18.5 kcal/mol for classical structures (Figure 6). Structures containing G-quadruplexes reached the highest stability of −72 kcal/mol (Figure 7).

## 3. Materials and Methods

### Data Processing

Data processing was performed using seven homemade Python scripts available as Appendix A. They must be executed in the following order: (1) search_stem_loop.py, (2) parse_uniprotdata.py, (3) analyze_stem_loop.py, (4) G_quartets.py, GO_terms.py, get_rnafold_structures.py. The Python module named toolbox.py is imported by some of these scripts (Appendix A). Coding DNA Sequences (CDSs) from genomic sequences of six eukaryotic and eight prokaryotic species were downloaded from the NCBI site ftp://ftp.ncbi.nlm.nih.gov/genomes/.

All CDSs from each species were first carefully checked in order to remove the ones that were non-nuclear or abnormal, like those annotated as pseudogenes. The RNAfold program from the ViennaRNA package 2 [51] was used to predict the minimum free energy (MFE) of secondary structures of RNA sequences located between the +16 and +65 positions, the A from the AUG initiator codon being numbered +1 in each CDS. Detection of G-Quadruplex structures was also incorporated into the RNAfold structure prediction algorithm. The number of guanines involved in G-Quadruplex structures was determined by counting the number of “+” characters in the bracket notation representing secondary structure printed by RNAfold. The MFE_struc_ calculated by RNAfold for a given structure was compared to the mean MFE_rand_ value obtained when the same sequence was randomized 100 times. The ratio MFE_struc_/MFE_rand_ allows to assess the reliability of the secondary structure prediction. The structures with a ratio MFE_struc_/MFE_rand_ equal or higher than 2 were selected. Importantly, the MFE value can be biased by the GC content of the RNA sequence. Therefore, the GC content was also calculated for each 50 nt RNA sequence submitted to RNAfold. PostScript files with plots of secondary structure graphs were produced by RNAfold for a few selected 50 nt CDS sequences. The functions of some CDSs are provided in the description lines of FASTA files downloaded from the NCBI site. Additional functional information can be obtained using the Gene Ontology (GO) resource. For each analyzed species, the relationship between a CDS and its associated GO terms was done parsing two text files, reviewed (Swiss-Prot database) and unreviewed (TrEMBL database), downloaded from the UniProt site https://www.uniprot.org/. We are aware that this selection approach is not fully exhaustive and might lead to loss of some putative interesting structures. A major drawback comes from the fact that randomization of sequences with a lot of repeated sequences is not expected to notably change the mfe. Indeed, these types of structures are most likely selected by our approach. Nevertheless, we decided to favor this approach that is more stringent than others in order to obtain a list of strong candidates rather than expanding our list by using alternative methods that might include false-positive structures. In other words, we choose to potentially omit false-positive structures rather than include in our selection false-negative structures that would appear with less stringent approaches.

## 4. Conclusions

Altogether, our analysis suggests that the START mechanism is putatively an important mechanism that is widely used in the three kingdoms of life to regulate translation of numerous mRNAs. The fact that RNA can modulate their folding in order to change their secondary structures on microsecond time scale [48,49] also suggests that the START mechanism can be used transiently for mRNA subclasses in specific conditions. The so-called “structuromes” of total RNA have been investigated both in vitro [52] and in vivo with DMS [53,54] and SHAPE [55]. Our study shed light on a novel feature indicating that a careful examination of putative secondary structures in the +16 to +65 window in the previously published genome-wide structural data might bring novel insights in the putative use of the START mechanism. In eukaryotes, it will be of particular interest to investigate the consensus sequences flanking the start codons of mRNAs that are potentially translated by the START mechanism. Since most of the eukaryotic start codons are embedded in the so-called Kozak sequence [56,57], the START mechanism possibly requires alternative consensus sequences around the start codon. In the future, the next step will be to confirm the implication of the selected structures in the START mechanism for efficient translation initiation. This can be done for instance by transplanting these selected structures in the +16 to +65 areas of reporter mRNAs in order to evaluate their impact on translation initiation efficiency. Finally, our study indicates that the START mechanism can be used in various prokaryotic and eukaryotic organisms. This suggests that the localization of the start codon by steric hindrance with downstream RNA secondary structures might be seen as an ancient mechanism that was maintained during evolution most probably because of its efficiency and simplicity. 

## Figures and Tables

**Figure 1 ijms-21-08373-f001:**
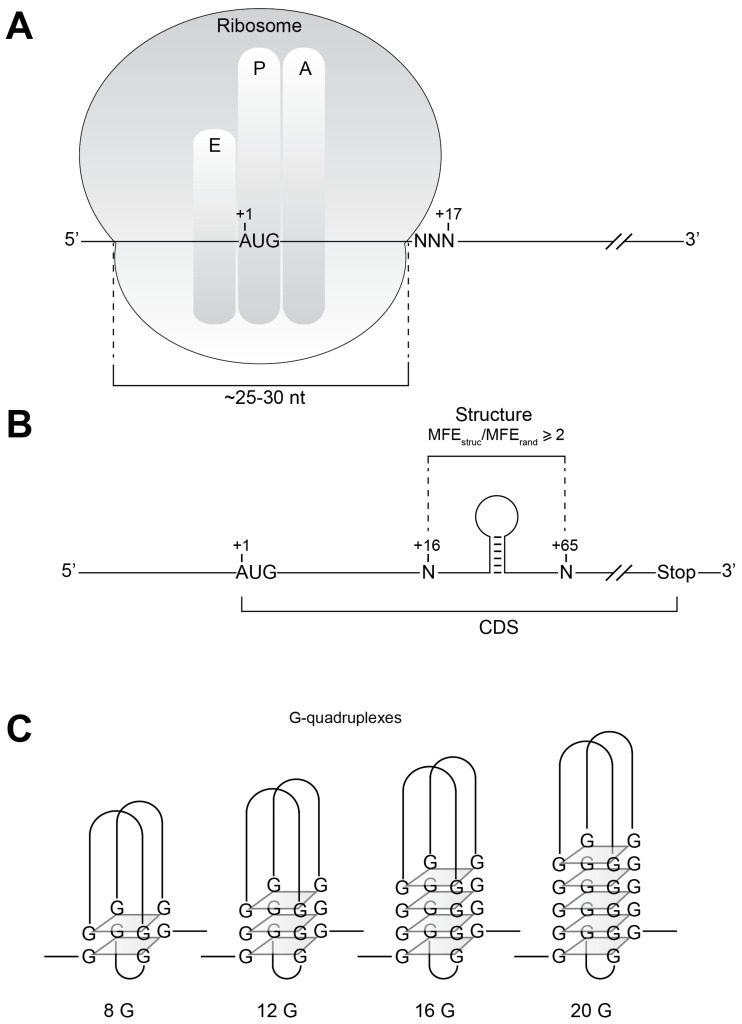
(**A**) During translation initiation, the assembled ribosome covers 15 to 17 nucleotides downstream from A (number +1 by convention) of the AUG start codon. In the Structure Assisted RNA Translation (START) mechanism, the AUG start codon is positioned in the P-site of the ribosome by stable secondary structures located immediately downstream of the +17 nucleotide. (**B**) In order to identify such putative new secondary structures, we screened the window +16 to +65 for putative secondary structures in the coding sequences (CDS). We focused on secondary structures with a minimal free energy (MFE_struc_) that is at least two times higher than the mfe of the same sequence that was randomized 100 times (MFE_rand_). (**C**) In addition, we also screened the same +16 to +65 window in CDS for G-quadruplex structures that contain 8, 12, 16 or 20 G-quartets.

**Figure 2 ijms-21-08373-f002:**
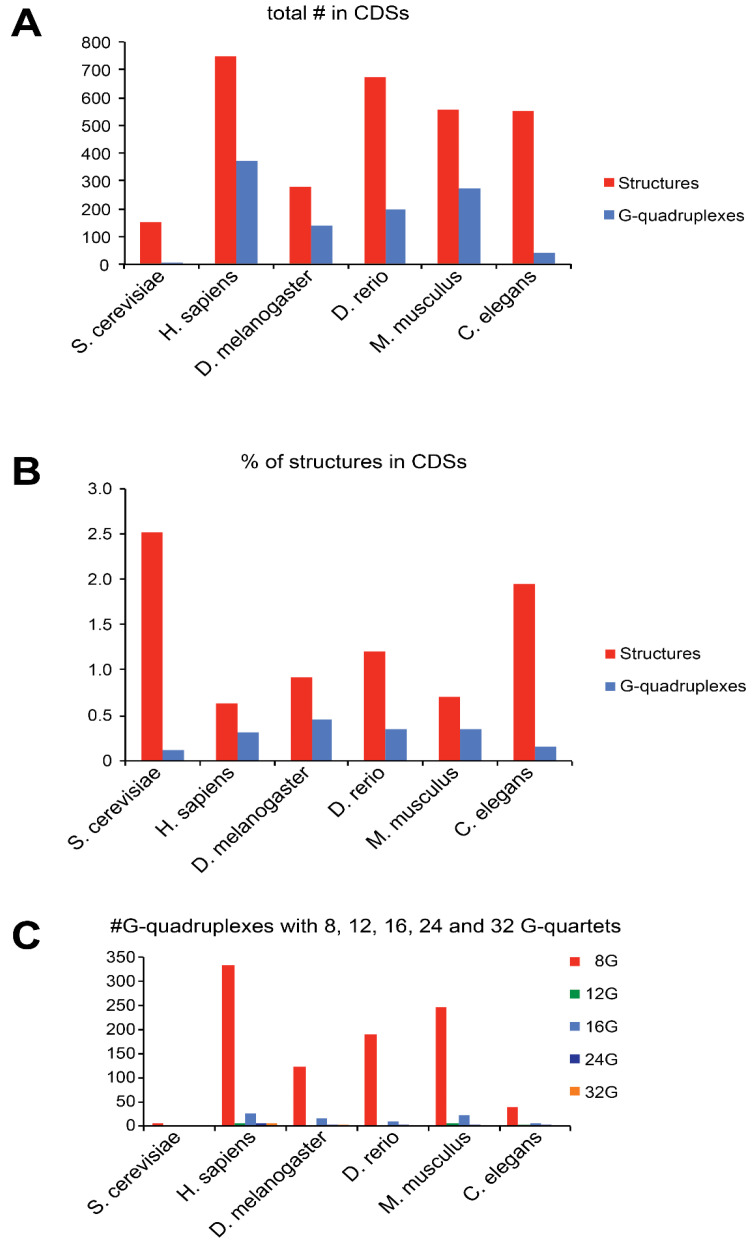
(**A**) The histogram represents the total number of putative secondary structures (red) and G-quadruplexes (blue) found in the +16 to +65 window of CDS that have a MFE_struc_/MFE_rand_ ratio of at least 2 in the CDSs from six eukaryotes. (**B**) The histogram represents the percentage of CDSs that contained secondary putative structures (red) and G-quadruplexes (blue). (**C**) The histogram represents the number of G-quadruplexes with 8, 12, 16, 24 and 32 G-residues in each organism.

**Figure 3 ijms-21-08373-f003:**
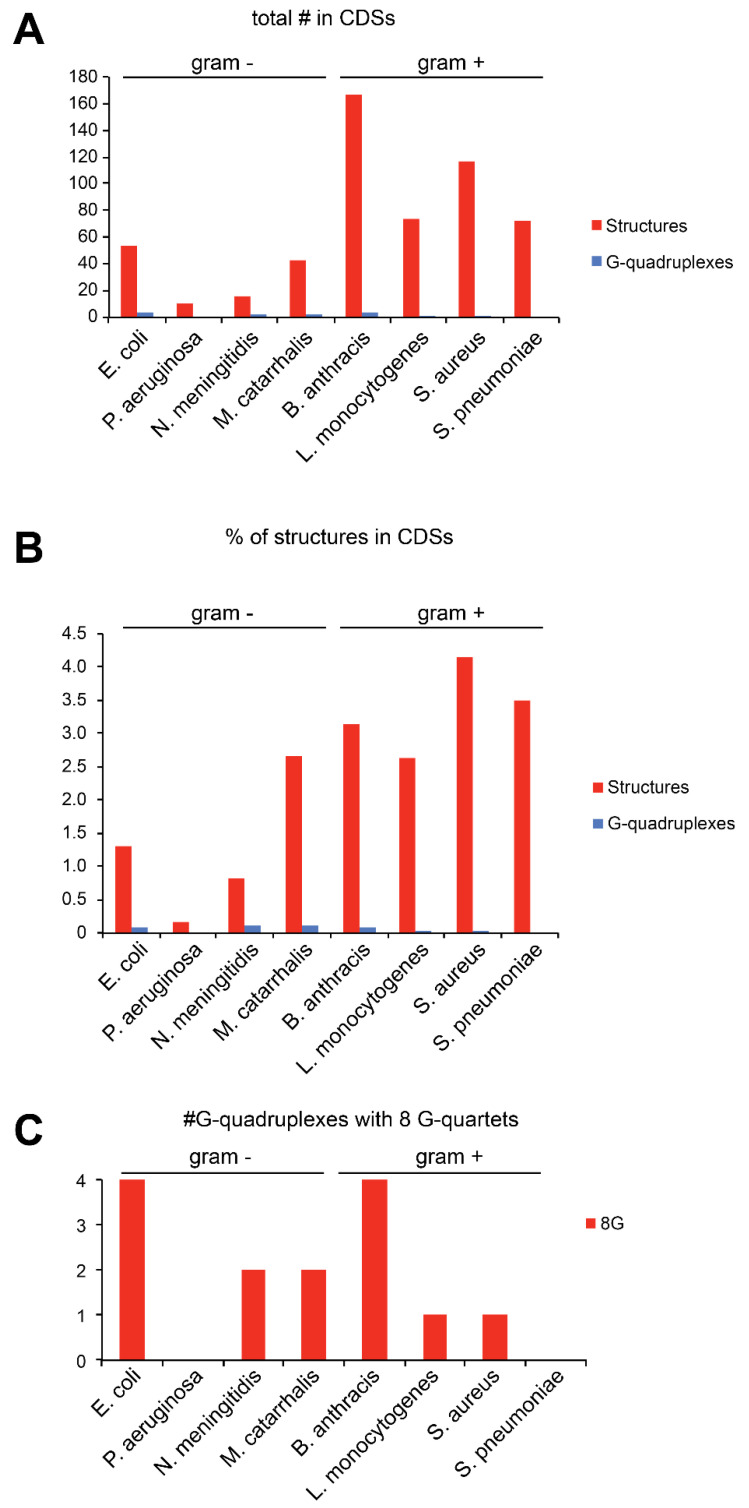
(**A**) The histogram represents the total number of putative secondary structures (red) and G-quadruplexes (blue) found in the +16 to +65 window of CDS that have a MFE_struc_/MFE_rand_ ratio of at least 2 in the CDSs of eight bacteria. Four Gram-negative bacteria are shown on the left and four Gram-positive are shown on the right. (**B**) The histogram represents the percentage of CDSs that contained secondary putative structures (red) and G-quadruplexes (blue). (**C**) The histogram represents the number of G-quadruplexes with 8 G residues in each organism.

**Figure 4 ijms-21-08373-f004:**
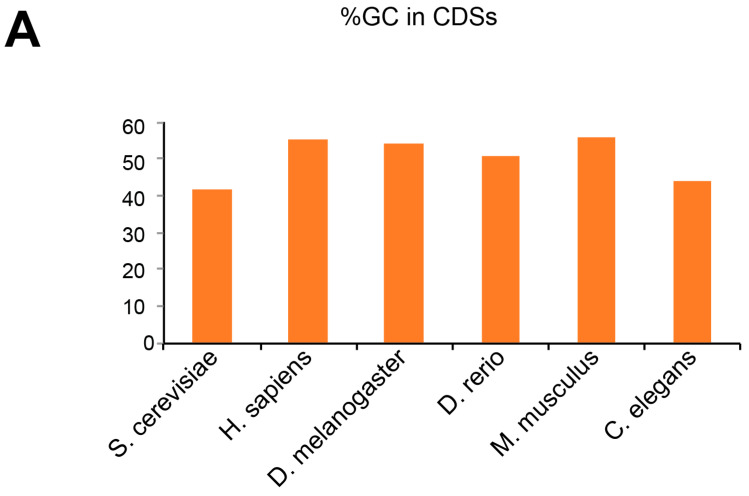
(**A**) The histogram represents the percentage of GC in the analyzed CDSs in eukaryotes. (**B**) The histogram represents the lowest minimal free energy of the top 25% of the structures found in the analyzed CDSs from eukaryotes.

**Figure 5 ijms-21-08373-f005:**
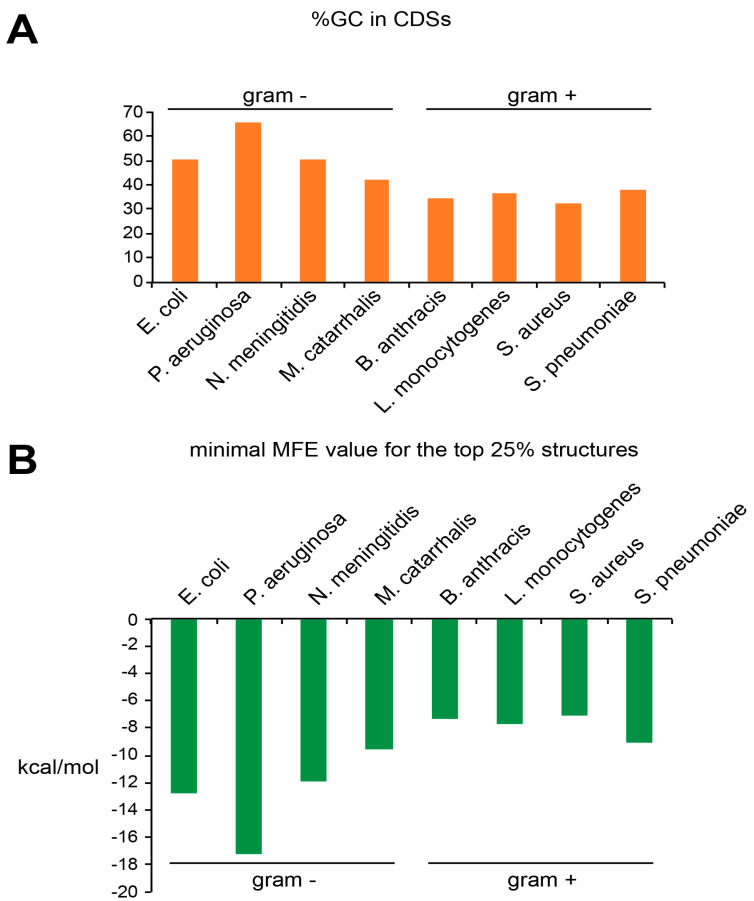
(**A**) The histogram represents the percentage of GC in the analyzed CDSs in Gram-negative and Gram-positive bacteria. (**B**) The histogram represents the lowest minimal free energy of the top 25% of the structures found the analyzed CDSs from Gram-negative and Gram-positive bacteria.

**Figure 6 ijms-21-08373-f006:**
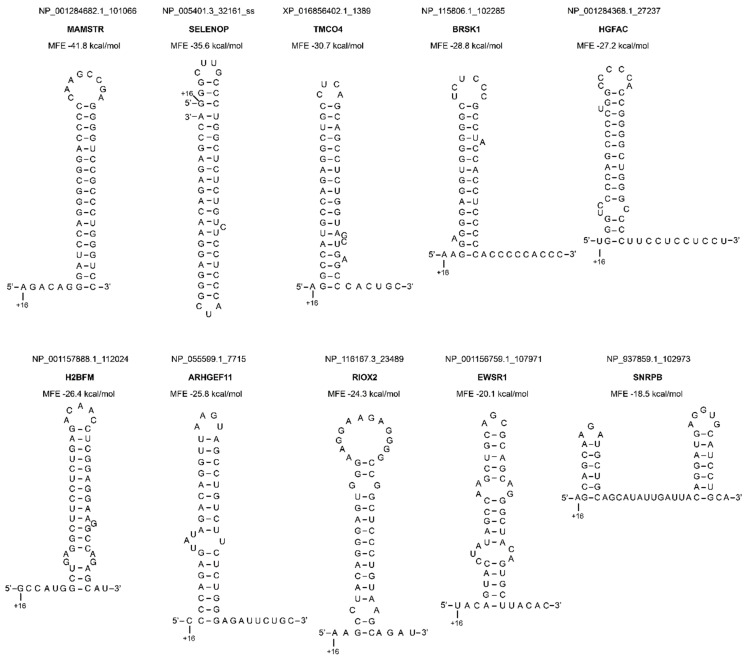
Representation of the ten best secondary structures selected in the human genome. The position of nucleotide +16 is indicated. The names of the genes are shown in bold.

**Figure 7 ijms-21-08373-f007:**
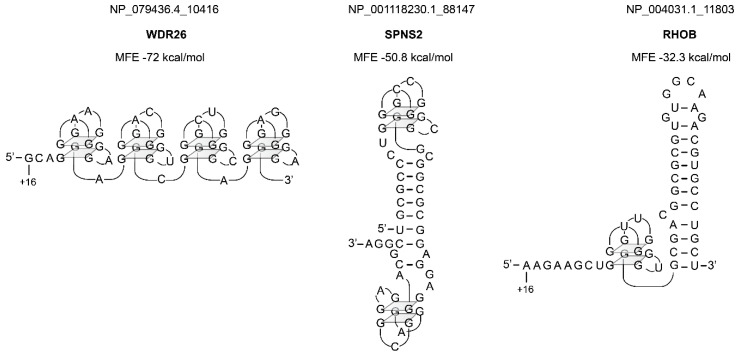
Representation of three structures containing G-quadruplexes selected in the human genome. The position of nucleotide +16 is indicated. The names of the genes are shown in bold.

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
