# Peer review of "How Many Messenger RNAs Can Be Translated by the START Mechanism?"

_ijms, 2020, doi:10.3390/ijms21218373_

Round 1
Reviewer 1 Report
The article by Despons L. and Martin F. aims at identifying in 6 eukaryotic and 8 bacterial genomes sequences forming stable secondary structure elements located downstream from the AUG start codon and possibly involved in the regulation of translation initiation efficiency. Because of their optimal positioning with respect to the AUG start codon, these sequences and structures would suggest a novel mechanism named the START mechanism for “Structure Assisted RNA translation” mechanism. The RNAfold program was used to predict Minimum Free Energy (MFE) of secondary structures located between the +16 and +65 positions (numbering starting at the A of the AUG codon). The MFEstructure calculated by RNAfold for a given structure was compared to the mean MFE value obtained after multiple randomization of the sequence (MFErand). The structures with a ratio MFEstructure/ MFErand ≥ 2 were selected and analyzed. The presence of putative G-quadruplex structures was also analyzed.
In human and mouse, G-quadruplexes were found in the selected sequences whereas the motif is rare in C. elegans and almost absent in S. cerevisiae. In bacteria, G-quadruplexes are very rare in the +16-+65 window. The number of selected sequences is higher in Gram-positive bacteria than in Gram-negative bacteria. Translation initiation mechanisms are less studied in Gram-positive bacteria. Therefore, the START mechanism might be an alternative to the mechanism involving Shine-Dalgarno sequences less frequently used in these organisms.
The influence of the GC content was then studied. In eukaryotes, the GC content is similar in all organisms and perfectly correlated to the stability of the structures. In bacteria, Gram-negatives have the highest GC content with the lowest number of structures selected. Therefore, the GC content can be a good predictor for “START” sequences in eukaryotes but not in bacteria.
Finally, the authors have analyzed the types of mRNAs containing these “START” sequences and found out that they could be particularly important for translation regulation in bacteria.
By studying 6 eukaryotic and 8 bacterial genomes, the authors have performed a wide study that suggests the participation of downstream stable secondary structures in translation initiation. In eukaryotes, G-quadruplex motifs might play an important regulatory function in these mechanisms. The article is concise, the data are well presented and well written and the article should be acceptable for publication.
Minor points:
-Prokaryotes should be replaced by bacteria, because in the present study, only bacterial genomes have been studied.
-line 13: participate in
-line 36: archaea
-line 81: identify
Author Response
Thank you for your comment ‘The article is concise, the data are well presented and well written’
Minor points:
-Prokaryotes should be replaced by bacteria, because in the present study, only bacterial genomes have been studied.
Corrected
-line 13: participate in
Corrected
-line 36: archaea
Corrected
-line 81: identify
Corrected

Reviewer 2 Report
Despons and Martin present a bioinformatic study predicting the occurrence of stable structures downstream of the annotated translation start sites, in the RNA of diverse species across domains of life. The structures may have a role in promoting recognition of the respective start codons by the initiating ribosomes or their small subunits. While the idea behind a stable secondary structure facilitating start codon recognition dates back to the origins of the understanding of translation mechanisms (i.e. Kozak’s works such as PMID 2601709, PMID 2236042, PMID 16213112), a comprehensive overview of the putative sites of such structures, which this work provides, can be of an interest to translation specialists and geneticists broadly. I think the manuscript can be cited in the future. However, there are several major conceptual as well as more technical concerns with the current version of the manuscript. Conceptually, in my opinion, the authors use a method for generating a “control” sequence that may not be valid for all possible nucleotide combinations, and can lead to omissions of cases, for instance, with less degenerate sequences which nonetheless would form a stable structure. The authors do not attempt to link their findings with any of “translation efficiency” estimates, now available for many of the species, or provide a comparison with the presence of a similarly-predicted structure directly over the start sites, which would have a negative effect. Further, while the authors mention structural probing of RNA in the end, they do not attempt to validate or discuss their findings in relation to these works. Such correlations and other indicated analyses would strengthen much the manuscript. Technically, the manuscript lacks several important references, explanations and discussions. For example, the entire concept of the stimulatory structure downstream of the start codon needs to be appropriately introduced and referenced, including references to the early works on translation. There needs to be more diversity in references covering mechanisms of translation initiation as well. The manuscript will benefit much from more discussions of the authors’ findings in the background of the existing literature, and strong summarising and concluding statements throughout all sections. Text flow needs to be streamlined. Given the overall interesting topic, my recommendation would be to re-consider the manuscript in IJMS after the above-mentioned concerns, and the more detailed points listed below, are addressed.
Abstract
-
Please consider making edits to produce a more concise text. i.e.:
“Translation initiation is a key step in general metabolism in all living cells.” → “Translation initiation is a key step in the protein synthesis stage of the gene expression pathway of all living cells.”.
“More precisely, this means that the ribosome has to find accurately the AUG start codon in order to ensure the integrity of the proteome.” → “In this important process, ribosomes have to accurately find the start codon in order to ensure the integrity of the proteome.”.
“The ‘Structure Assisted RNA Translation’ mechanism or ‘START’ is using stable secondary structures located in the coding sequence that participate to the AUG selection by steric hindrance of the pre-initiation complex.” → “‘Structure Assisted RNA Translation’ mechanism, or ‘START’, has been proposed to use stable secondary structures located in the coding sequence to augment start site selection by steric hindrance of the progression of pre-initiation complex on messenger RNA.”.
-
Structure Assisted RNA Translation needs to be introduced better.
-
“in a window spanning the nucleotides from +16 to +65” – please provide at least a minimal justification of the selected window/region. It is mentioned later in the Results and Discussion but still needs to be appropriately introduced here.
-
“half of them are G-quadruplex structures” – are these predicted or verified G-quadruplexes? Maybe to include a brief statement on this.
-
Abstract could benefit from a concluding sentence or two. What is the importance and novelty of the finding? Any evolutionary implications or those related to the ‘lifestyle’ differences between the compared domains of life?
Introduction
-
“Among these, the rate-limiting step is translation initiation.” – I agree that initiation control is often predominant and is very important, allowing an ease of per-CDS protein output adjustment. However, there are multiple instances where, at least, elongation rate control can dominate initiation rate control. Thus, the statement needs to be justified and re-written to include these caveats and references.
-
“generally an AUG codon” → “most frequently encoded by an AUG triplet”.
-
References 1-5 – generally a good selection but maybe to expand it with PMID 32869519, PMID 32575790, PMID 32518156, PMID 31220979, PMID 29735639, PMID 29624880, at least, just to provide a slightly more comprehensive overview – I understand the topic is vast and it would be impossible to cite everything. Nonetheless, it is important to provide a diversity of opinions.
-
“no structure should be present in the vicinity of the AUG” – while it is relatively clear to me what the authors mean, this statement can be confusing for may readers, especially in the background of the subsequent material. Would not it be better to introduce the near-start-site structural problem somewhat more in the detail early on and cite the respective works? The accepted notion in the field, at least for eukaryotic initiation where it is important to prevent pre-initiation complex ‘leakage’, is that any structure directly at or before (upstream of) a start codon is inhibitory, whereas a relatively immediate structure downstream of it can be stimulatory. However, there are obvious exceptions in the form of Internal Ribosome Entry Sites even from this rule – these are often extremely strongly structured and the structure overlaps with the start site. All this needs to be introduced and discussed.
-
“Shine Dalgarno” → “Shine-Dalgarno” or, less often, “Shine and Dalgarno”.
-
“encompasses the ‘Shine Dalgarno’ (SD)” – please briefly explain the SD-dependent and SD-independent mechanisms of prokaryotic initiation. This can be difficult to comprehend for a reader not immediately acquaint with translation (initiation).
-
“Nowadays this point is very well documented.” - please include references supporting this statement (there are plenty of, including reviews).
-
Lines 63-64 – yet, there are several proposals about the potential mechanisms. For example, those based on S1 small ribosomal subunit protein functionality. It would be good to briefly mention/discuss these mechanisms.
-
“In eukaryotes, the 5’UTR are generally much longer” – unclear to what this comparison is made. It would be good to support it with numbers and references.
-
“eIF1a” → “eIF1A” according to the field-accepted conventions. Please make the corresponding changes throughout.
-
Line 68, please unfold “ternary complex”. Also, other factors are likely present (at least, in a “general case”), such as eIF4F and its components, eIF4B likely.
-
“the AUG start codon” – I would strongly suggest to make these statements more general, such as “the start codon”. Non-AUG start codons are sometimes the preferred ones on some mRNAs.
-
“of mRNAs is most of the time”→ “of mRNAs is, for the mot of time,”.
-
In similarity to the Abstract, main findings and conclusions are missing in the end of the Introduction.
Results and Discussion
-
Lines 118-120 “covers a region encompassing approximately 25 to 30 nucleotides” – please include the supporting references.
-
Lines 140-148 – generally I like the approach, although it probably is difficult to fully justify the particular method of control signal generation though randomisation. Could the authors discus and introduce it more? What was the decision-making process behind selecting this approach? Were any alternatives considered? My main concern is that the approach can potentially not work well for insufficiently degenerate sequences. For example, sequences with over-represented Watson-Crick complementary and non-Watson-Crick pairing bases can give false negatives with this approach. I.e., for a simple reference, what will happen with a sequence that is 80% or so C,G-comprised? It needs to be at least estimated what is the proportion of these sequences; better yet, an alternative approach used for these.
-
Line 166: “G-quadruplexes are pretty rare” → “G-quadruplexes are rare”.
-
“bacterias” → “bacteria”.
-
Line 182: “G-quadruplexes are pretty rare”→ “G-quadruplexes are rare”.
-
Figure 5A, legend, line 246 – double dot separated by a space in the end of the legend.
-
Line 255 and further, Figure 6, Figure 7 – inconsistent use of the energy units. Probably, the most accepted are “kcal/mol”, “kcal×mol-1”, “kcal·mol-1”, not with a full stop or space.
-
Figure 6, Figure 7, and the corresponding text, please indicate and justify exactly how the free energy was predicted. Calculations can vary widely depending on the model and parameters used.
-
Lines 169, 170 – the authors mention transcriptome-wide structural measurements. These now exist in multiple variants, for yeast and higher eukaryotic RNA as well as for RNA of some prokaryotes. Are there any parallels observed in these studies that would additionally validate the authors’ findings? Even if not, maybe to discuss it and provide an explanation? These additions can strengthen the manuscript. Currently, this analysis is lacking and the corresponding discussion is too brief and under-developed.
-
Manuscript needs to have a better conceptualised, presented and written concluding part in the end of the Results and Discussion section.
-
In Supplementary Table 1 – please use field-accepted spelling for Eukaryotes and Prokaryotes.
Author Response
Reviewer 2
Abstract
- Please consider making edits to produce a more concise text. i.e.:
“Translation initiation is a key step in general metabolism in all living cells.” → “Translation initiation is a key step in the protein synthesis stage of the gene expression pathway of all living cells.”.
“More precisely, this means that the ribosome has to find accurately the AUG start codon in order to ensure the integrity of the proteome.” → “In this important process, ribosomes have to accurately find the start codon in order to ensure the integrity of the proteome.”.
“The ‘Structure Assisted RNA Translation’ mechanism or ‘START’ is using stable secondary structures located in the coding sequence that participate to the AUG selection by steric hindrance of the pre-initiation complex.” → “‘Structure Assisted RNA Translation’ mechanism, or ‘START’, has been proposed to use stable secondary structures located in the coding sequence to augment start site selection by steric hindrance of the progression of pre-initiation complex on messenger RNA.”
- Structure Assisted RNA Translation needs to be introduced better.
- “in a window spanning the nucleotides from +16 to +65” – please provide at least a minimal justification of the selected window/region. It is mentioned later in the Results and Discussion but still needs to be appropriately introduced here.
- “half of them are G-quadruplex structures” – are these predicted or verified G-quadruplexes? Maybe to include a brief statement on this.
- Abstract could benefit from a concluding sentence or two. What is the importance and novelty of the finding? Any evolutionary implications or those related to the ‘lifestyle’ differences between the compared domains of life?
The abstract has been substantially remodelled according to the reviewer’s suggestions.
Introduction
- “Among these, the rate-limiting step is translation initiation.” – I agree that initiation control is often predominant and is very important, allowing an ease of per-CDS protein output adjustment. However, there are multiple instances where, at least, elongation rate control can dominate initiation rate control. Thus, the statement needs to be justified and re-written to include these caveats and references.
We added the following sentences to address this point: lines 35-39
Among these, the rate-limiting step is most of the time translation initiation and is therefore considered as the master regulator step, although there are examples where the elongation rate can dominate initiation rate control. In this initiation phase, the ribosome has to localise accurately the start codon, most frequently encoded by an AUG triplet.
- “generally an AUG codon” → “most frequently encoded by an AUG triplet”.
As requested by the reviewer, we replaced the word ‘codon’ by ‘triplet’; See also point 6.
- References 1-5 – generally a good selection but maybe to expand it with PMID 32869519, PMID 32575790, PMID 32518156, PMID 31220979, PMID 29735639, PMID 29624880, at least, just to provide a slightly more comprehensive overview – I understand the topic is vast and it would be impossible to cite everything. Nonetheless, it is important to provide a diversity of opinions.
All the 6 requested references have been inserted in the revised manuscript. We thank the reviewer for these constructive suggestions, however we would like to emphasize that 3 references among the suggested ones were published after June 2020, since our manuscript was submitted in March, we could obviously not insert them at that time.
- “no structure should be present in the vicinity of the AUG” – while it is relatively clear to me what the authors mean, this statement can be confusing for may readers, especially in the background of the subsequent material. Would not it be better to introduce the near-start-site structural problem somewhat more in the detail early on and cite the respective works? The accepted notion in the field, at least for eukaryotic initiation where it is important to prevent pre-initiation complex ‘leakage’, is that any structure directly at or before (upstream of) a start codon is inhibitory, whereas a relatively immediate structure downstream of it can be stimulatory. However, there are obvious exceptions in the form of Internal Ribosome Entry Sites even from this rule – these are often extremely strongly structured and the structure overlaps with the start site. All this needs to be introduced and discussed.
We added the following sentences and 1 reference to address this point: lines 68-71.
In addition, it is well established that structures directly at or in the close vicinity of a start codon are inhibitory because they prevent efficient ribosome loading [10]. However, in the case of Internal Ribosome Entry Sites (IRES), structures located close to the start codons are usually crucial for efficient translation (Mailliot & Martin, 2018).
- “Shine Dalgarno” → “Shine-Dalgarno” or, less often, “Shine and Dalgarno”.
We used ‘Shine-Dalgarno’
- “encompasses the ‘Shine Dalgarno’ (SD)” – please briefly explain the SD-dependent and SD-independent mechanisms of prokaryotic initiation. This can be difficult to comprehend for a reader not immediately acquaint with translation (initiation).
We added the following sentence to address this point: lines 74-75.
During initiation, the 3’ end of the 16S ribosomal RNA anneals with the SD sequence on the mRNA in order to ensure efficient assembly of the pre-initiation complex.
- “Nowadays this point is very well documented.” - please include references supporting this statement (there are plenty of, including reviews).
We added the following sentence and a reference to address this point: lines 84-85.
Nowadays this point is very well documented (reference 1 and references therein).
- Lines 63-64 – yet, there are several proposals about the potential mechanisms. For example, those based on S1 small ribosomal subunit protein functionality. It would be good to briefly mention/discuss these mechanisms.
We added the following sentences and 2 references to address this point: lines 79-80.
The largest acidic ribosomal protein S1 is required for efficient unfolding of RNA duplexes hindering the SD sequence (Duval et al., 2013 ; Byrgazov et al., 2015)
- “In eukaryotes, the 5’UTR are generally much longer” – unclear to what this comparison is made. It would be good to support it with numbers and references.
We added the following sentences and 5 references to address this point: lines 94-95.
The median length of 5’UTR in eukaryotes ranges from 53 nucleotides in yeast to 218 nucleotides in humans.
(Leppek et al., 2018; Pesole et al., 2001, Lynch et al., 2005, Mignone et al., 2002, Jan et al., 2011)
- “eIF1a” → “eIF1A” according to the field-accepted conventions. Please make the corresponding changes throughout.
Corrected.
- Line 68, please unfold “ternary complex”. Also, other factors are likely present (at least, in a “general case”), such as eIF4F and its components, eIF4B likely.
We added the following sentences to address this point: lines 95-101.
First, an mRNA activation step triggers the assembly of the eIF4F complex on the 5’ cap. This complex contains eIF4E, the cap binding protein, the RNA helicase eIF4A and its auxiliary factor eIF4B and the platform protein eIF4G. Then, the assembly of the translation machinery occurs at the 5’ cap and a so-called scanning step enables the 5’-3’ sliding of the 43S particle (which contains the 40S ribosomal subunit, the multi subunit factor eIF3, the scanning factors eIF1, 1A and 5 and the so-called ternary complex (comprising eIF2 and the initiator tRNAMet)) in order to localise the start codon.
- “the AUG start codon” – I would strongly suggest to make these statements more general, such as “the start codon”. Non-AUG start codons are sometimes the preferred ones on some mRNAs.
Corrected.
- “of mRNAs is most of the time”→ “of mRNAs is, for the mot of time,”.
Corrected.
- In similarity to the Abstract, main findings and conclusions are missing in the end of the Introduction.
Results and Discussion
We added the following sentences to address this point: lines 125-131.
More precisely, the START mechanism relies on the presence of stable secondary structures located in the coding region. These structures are located at the appropriate distance from the start codon (+16 to +19) that enables the assembly of the pre-initiation complex on the AUG start codon without unfolding of the structures. In fact, these structures at this position slow down by steric hindrance the progression of the scanning pre-initiation and thereby augment the initiation efficiency. Thus, their precise position downstream +16 to +19 is crucial for the efficient localisation of the start codon.
- Lines 118-120 “covers a region encompassing approximately 25 to 30 nucleotides” – please include the supporting references.
We added 2 references to address this point: line 169.
(Steitz, 1969) and (Wolin & Walter, 1988).
- Lines 140-148 – generally I like the approach, although it probably is difficult to fully justify the particular method of control signal generation though randomisation. Could the authors discus and introduce it more? What was the decision-making process behind selecting this approach? Were any alternatives considered? My main concern is that the approach can potentially not work well for insufficiently degenerate sequences. For example, sequences with over-represented Watson-Crick complementary and non-Watson-Crick pairing bases can give false negatives with this approach. I.e., for a simple reference, what will happen with a sequence that is 80% or so C,G-comprised? It needs to be at least estimated what is the proportion of these sequences; better yet, an alternative approach used for these.
The method we have chosen for structure selection is clearly explained in the material and method section (lines 137-166). We are aware that this selection approach is not fully exhaustive and might lead to lose some putative interesting structures. For instance, randomization of sequences with a lot of repeated sequences is not expected to change a lot the mfe. Indeed, these types of structures are not selected by our approach. Nevertheless, we decided to favour our approach that is more stringent than others in order to obtain a list of strong candidates rather than expanding our list by using alternative methods that might include false positive structures. In other words, we choose to potentially omit false-positive structures rather than include in our selection false-negative structures that would appear with less stringent approaches.
- Line 166: “G-quadruplexes are pretty rare” → “G-quadruplexes are rare”.
Corrected.
- “bacterias” → “bacteria”.
Corrected.
- Line 182: “G-quadruplexes are pretty rare”→ “G-quadruplexes are rare”.
Corrected.
- Figure 5A, legend, line 246 – double dot separated by a space in the end of the legend.
Corrected.
- Line 255 and further, Figure 6, Figure 7 – inconsistent use of the energy units. Probably, the most accepted are “kcal/mol”, “kcal×mol-1”, “kcal·mol-1”, not with a full stop or space.
We used ‘kcal/mol’
Figure 6 and 7 have been modified accordingly.
- Figure 6, Figure 7, and the corresponding text, please indicate and justify exactly how the free energy was predicted. Calculations can vary widely depending on the model and parameters used.
The reviewer is right, there are different ways for mfe calculation and not prediction. In our case, as is is clearly mentioned in the ‘material and method section’, we used the Vienna RNA package 2.4.14. This package includes the RNAfold algorithm, RNA secondary predictions and mfe calculations are done by this well-renowed RNAfold algorithm, in our submitted manuscript we already referred to the original publication (Lorenz et al., 2011). (Lines 145-147).
- Lines 169, 170 – the authors mention transcriptome-wide structural measurements. These now exist in multiple variants, for yeast and higher eukaryotic RNA as well as for RNA of some prokaryotes. Are there any parallels observed in these studies that would additionally validate the authors’ findings? Even if not, maybe to discuss it and provide an explanation? These additions can strengthen the manuscript. Currently, this analysis is lacking and the corresponding discussion is too brief and under-developed.
We have already addressed this point in the two previous revisions of our manuscript. The tools to compare our predicted structures with published structuromes do not exist. To be exhaustive, one would need to compare our structures with all the published data and not only a specific publication that would lead to a biased analysis. Such an exhaustive comparison would require an enormous amount of work that cannot be done in this time frame.
- Manuscript needs to have a better conceptualised, presented and written concluding part in the end of the Results and Discussion section.
We have added the concluding sentences to address this point: lines 365-369
Finally, our study indicates that the START mechanism is used in various prokaryotic and eukaryotic organisms. This suggests that the localization of the start codon by steric hindrance with downstream RNA secondary structures might be seen as an ancient mechanism that was maintained during evolution most probably because of its efficiency and simplicity.
- In Supplementary Table 1 – please use field-accepted spelling for Eukaryotes and Prokaryotes.
Corrected.

Round 2
Reviewer 2 Report
The authors have improved the manuscript. Nonetheless, I believe that some corrections are necessary to make it acceptable for IJMS. I do not wish to block the manuscript as I find the idea and information as potentially useful and interesting. However, my major concern is the pitch of the description of the results and the remaining discrepancy between the data shown and the conclusions. Below is the summary of my key concerns. I think these could be partly addressed by re-writing and moderating the statements.
1. In the title (and overall), the authors need to add a tentative mode. Something like “How many messenger RNAs can be translated by the START mechanism?” It does not diminish the findings, but describes the work more precisely, which in turn creates more confidence. To answer how many mRNAs are translated via START, the authors would need to: (a) provide the proof of the existence of the corresponding structures in the mRNAs in vivo or under the conditions of translation, and (b) provide the proof that upon a careful disruption of these structures, the initiation frequency is decreased over the corresponding start sites (and ideally increased on a downstream start site, at least in the case of eukaryotes). The authors indicated in the response that both tasks are outside the scope of the work. I can agree with this statement. But then the conclusions, discussions and the title need to be modified accordingly.
2. I can agree with the authors’ argumentation in response to my point 21, about the method of “baseline” calculation through iterative randomisation (lines 151-154 of the revised text). However, logic incorporated into the authors’ response to my comments needs to be transposed into the Methods section of the manuscript. It explains the pros and cons of the approach and the limitations, thus adding value to the work. It would be ideal to support the approach with cited works that use similar methods, too.
3. (applies to all similar statements) In the end of Results and Discussion of the revised manuscript, please change “Finally, our study indicates that the START mechanism is used in various prokaryotic and eukaryotic organisms.” into a suggestive phrasing such as “Finally, our study indicates that the START mechanism can be used in various prokaryotic and eukaryotic organisms.”.
4. Minor – the additional references included in the revision are commented “to insert” but the actual manuscript text has not been changed in this regard.
Author Response
- In the title (and overall), the authors need to add a tentative mode. Something like “How many messenger RNAs can be translated by the START mechanism?” It does not diminish the findings, but describes the work more precisely, which in turn creates more confidence. To answer how many mRNAs are translated via START, the authors would need to: (a) provide the proof of the existence of the corresponding structures in the mRNAs in vivo or under the conditions of translation, and (b) provide the proof that upon a careful disruption of these structures, the initiation frequency is decreased over the corresponding start sites (and ideally increased on a downstream start site, at least in the case of eukaryotes). The authors indicated in the response that both tasks are outside the scope of the work. I can agree with this statement. But then the conclusions, discussions and the title need to be modified accordingly.
We have modified our manuscript according to the reviewer’s suggestions. They are highlighted in yellow.
- I can agree with the authors’ argumentation in response to my point 21, about the method of “baseline” calculation through iterative randomisation (lines 151-154 of the revised text). However, logic incorporated into the authors’ response to my comments needs to be transposed into the Methods section of the manuscript. It explains the pros and cons of the approach and the limitations, thus adding value to the work. It would be ideal to support the approach with cited works that use similar methods, too.
We have added the following sentences to the methods section. They are highlighted in yellow (lines 166-174).
We are aware that this selection approach is not fully exhaustive and might lead to lose some putative interesting structures. A major drawback comes from the fact that randomization of sequences with a lot of repeated sequences is not expected to change a lot the mfe. Indeed, these types of structures are most likely selected by our approach. Nevertheless, we decided to favour this approach that is more stringent than others in order to obtain a list of strong candidates rather than expanding our list by using alternative methods that might include false positive structures. In other words, we choose to potentially omit false-positive structures rather than include in our selection false-negative structures that would appear with less stringent approaches.
- (applies to all similar statements) In the end of Results and Discussion of the revised manuscript, please change “Finally, our study indicates that the START mechanism is used in various prokaryotic and eukaryotic organisms.” into a suggestive phrasing such as “Finally, our study indicates that the START mechanism can be used in various prokaryotic and eukaryotic organisms.”.
Corrected.
- Minor – the additional references included in the revision are commented “to insert” but the actual manuscript text has not been changed in this regard.
References inserted.
This manuscript is a resubmission of an earlier submission. The following is a list of the peer review reports and author responses from that submission.
Round 1
Reviewer 1 Report
The manuscript is overall well written and gives a clear overview of the START mechanism for eucaryotes and bacteria. It shows how regions downstream of the AUG start codon can help in translation initiation.
General remarks :
- All the figures are poorly resolved. Please enhance the resolution of your images.
- The format for the references is not the same for all. Please uniform.
- The method starting line 135 should be in the method section
Specific remarks :
Line 67 : for a general reader, the term "43S particle" has not been described previously in the text. Please explain briefly in parenthesis what is a 43S particle.
line 162. C. elegans : with a space between . and elegans
line 167 : the legend doesn’t correspond with the figures in Figure2. Figure 2C in missing
line 179 : there is a space needed between the gender and the species
line 190 : same thing with the figure and the legend. Figure 3C is missing
line 200 : might be
line 224 : 25% of the structures found in the…
line 227 : : there is a space needed between the gender and the species
line 231 : : there is a space needed between the gender and the species
Author Response
Reviewer1
General remarks :
- All the figures are poorly resolved. Please enhance the resolution of your images.
Done
- The format for the references is not the same for all. Please uniform.
Done
- The method starting line 135 should be in the method section
We have described the experimental strategy in the method section. However, we believe that it’s important for the reader to briefly describe our approach also at the beginning of the results section to improve the clarity of the manuscript. Therefore, we would prefer to keep this part here.
Specific remarks :
Line 67 : for a general reader, the term "43S particle" has not been described previously in the text. Please explain briefly in parenthesis what is a 43S particle.
Done
line 162. C. elegans : with a space between . and elegans
Done
line 167 : the legend doesn’t correspond with the figures in Figure2. Figure 2C in missing
We have inserted the missing histogram.
line 179 : there is a space needed between the gender and the species
Done
line 190 : same thing with the figure and the legend. Figure 3C is missing
We have inserted the missing histogram.
line 200 : might be
Done
line 224 : 25% of the structures found in the…
Done
line 227 : : there is a space needed between the gender and the species
Done
line 231 : : there is a space needed between the gender and the species
Done
Reviewer 2 Report
Title: How many messenger RNAs are translated by the START mechanism?
The authors have previously published a hypothetical paper on START mechanism in 2018
(START: STructure-Assisted RNA Translation, RNA Biol. 2018; 15(9): 1250–1253).
In this paper they claim to have developed a "bioinformatic tool to screen coding sequences" for structures compatible with their previously proposed mechanism (START).
While the idea is interesting and the hypothesis may be correct, they do not add more information to their 2018 paper, and they don't describe or present the tool they developed.
Throughout the paper they talk about 2 kingdom of life but they claim their hypothesis holds in 3 kingdom of life without any more information/justification.
It is not clear why they chose the structures with MFE_struc/MFE_rand >2. Since the MFE_rand is created by randomly shuffling the original sequence (about 50 bp long), when the sequence has a lot of repeats then they should not expect the shuffle to change the MFE of the random sequence much. These are some possible cases that are not considered in this work while might be potentially interesting structures.
In addition the 50 bp area they consider might be involved with long range interactions (with the flanking sequences). Therefore, the local structure of the short sequence they find in this area may not be part of the global structure.
They could have used available experimental data on the structure in those regions to provide a more reliable information on local structure.
Alternatively, their START mechanism could be backed up by structural conservation found by structural alignment.
The paper is not well written. There are multiple typos in the text (see for example line 220). The figures are of low quality. Caption of Figures 2 and 3, refer to subfigure (C) which does not exist in any of the two.
The use for Figure 6 is not clear. If these structures are the only or consensus structures of the START mechanism, then by all means adding them to the paper would add value, but right now they are presented as sample structures and so don't add value.
The authors claim that G-quadruplexes are important structures in START mechanism but do not add more information as why they might be important.
Overall I think they should address the following points before the paper can be published.
1. Is there any biological verification of this START phenomenon or is this all data that supports the possibility of it? Their 2018 RNA biology paper sounds hypothetical as well.
2. They rightly point out Shine-Dalgarno in prokaryotes but do not talk about Kozak sequences in eukaryotes at all. While the ribosome does mount the mRNA at the 5' G cap, it slides along until it finds the Kozak sequence to begin translation initiation. It would be interesting to see the co-occurrence of Kozak sequence with their START structures.
3. They talk about prokaryotic transcription but they don’t talk about Internal Ribosome Entry Sites at all. Prokaryotic transcripts are typically polycistronic, with translation of multiple embedded sequences from the same transcript possible at the same time. Do IRES sites also show START structures?
4. From line 242 they talk about gene ontology (GO) annotation. This is useful data but we only have annotations for things that have been observed or predicted. This isn’t all of the functions for a gene. It also includes information about pathological features. I would be very cautious about using the GO annotations as a be all, end all as to gene function. Additionally, GO terms include predicted, not verified function and localization. So they may be talking about predicted functions for transcripts with predicted secondary structural regulation
Author Response
Reviewer 2
Overall I think they should address the following points before the paper can be published.
1. Is there any biological verification of this START phenomenon or is this all data that supports the possibility of it? Their 2018 RNA biology paper sounds hypothetical as well.
This statement is not correct. Our 2018 RNA biology paper has indeed reported only validated examples of the START mechanism. All these examples described in this publication are actually supported by solid functional data and for some cases even structural data (like for histone H4 for example). This was precisely the goal of this seminal publication on the START mechanism published in 2018 in RNA Biology. The present manuscript is a follow-up of this paper. International Journal of Molecular Sciences invited us to write this follow-up for a special issue entitled ‘RNA structure prediction’. The goal of our manuscript is to open new avenues for further biological validations of new examples of messenger RNA that could be potentially translated by the START mechanism, therefore we believe that our manuscript is perfectly in the scope of this special issue.
They rightly point out Shine-Dalgarno in prokaryotes but do not talk about Kozak sequences in eukaryotes at all. While the ribosome does mount the mRNA at the 5' G cap, it slides along until it finds the Kozak sequence to begin translation initiation. It would be interesting to see the co-occurrence of Kozak sequence with their START structures.
We gratefully appreciate this comment. We have added the following sentences and two references in the ‘discussion’ section.
‘In eukaryotes, it will be of particular interest to investigate the consensus sequences flanking the start codons of mRNAs that are potentially translated by the START mechanism. Since most of the eukaryotic start codons are embedded in the so-called Kozak sequence [37,38], the START mechanism possibly requires alternative consensus sequences around the start codon.’
We also want to emphasize that our selection was made in the window +16-+65 window. We performed our selection on Coding Sequences databases that have been carefully cleaned from pseudo-genes. The beginning of the sequences that were used for our screenings is the start codon, AUG or AUG-like codons. The Kozak consensus sequences is flanking the start codon, especially nucleotide upstream the start codon at position -3 is critical. Therefore retrieving the AUG flanking sequences would require a significant amount of work that cannot be done in the time given for the revision (10 days). Although, we acknowledge that this is an important question, we believe that this point on the Kozak sequence is beyond the scope of this manuscript and will be addressed in our next publication(s) on the START mechanism.
- They talk about prokaryotic transcription but they don’t talk about Internal Ribosome Entry Sites at all. Prokaryotic transcripts are typically polycistronic, with translation of multiple embedded sequences from the same transcript possible at the same time. Do IRES sites also show START structures?
IRES elements are most of the time located in the 5’UTR and therefore upstream of the AUG start codon. Therefore, for the same reason as detailed in point 2, we cannot assess this question in this timing. Moreover, cellular IRES are poorly defined and it’s very difficult to predict them since they do not share obvious common RNA motifs. Here again, we acknowledge the importance of this question but we will address it in our next publication(s) but not in this manuscript.
From line 242 they talk about gene ontology (GO) annotation. This is useful data but we only have annotations for things that have been observed or predicted. This isn’t all of the functions for a gene. It also includes information about pathological features. I would be very cautious about using the GO annotations as a be all, end all as to gene function. Additionally, GO terms include predicted, not verified function and localization. So they may be talking about predicted functions for transcripts with predicted secondary structural regulation
We gratefully appreciate this comment. We have modified our data in order to discriminate predicted GO terms from experimentally validated GO terms. We discriminated three catagories of GO terms (U: unreviewed, R: reviewed, unk: unknown) in new versions of supplemental tables 2 and 3. This point now appears clearly in the corrected presented data in the supplemental tables 2 and 3.